

# Combining the U-Net model and a Multi-textRG algorithm for fine SAR ice-water classification

Yan Sun[1,2], Shaoyin Wang[1,2], Xiao Cheng[1,2], Teng Li[1,2], Chong Liu[1,2], Yufang Ye[1,2], Xi Zhao[1,2]

[1]School of Geospatial Engineering and Science, Sun Yat-sen University, Guangzhou, China
[2]Southern Marine Science and Engineering Guangdong Laboratory and the Key Laboratory of Comprehensive Observation of Polar Environment, Ministry of Education, Sun Yat-sen University, Zhuhai, China

*Correspondence to*: Xiao Cheng (chengxiao9@mail.sysu.edu.cn)

**Abstract.** Sea ice classification faces challenges due to the similarity among surfaces such as wind-driven open water (OW), smooth thin ice, level first-year ice (FYI), and melted ice surfaces. Previous algorithms combine unsupervised region
segmentation and supervised neural networks, yet struggle due to limited manual labels and inaccurate region segmentation. We propose to adopt a supervised neural network followed by a region segmentation algorithm with experiential knowledge involved to solve the ambiguous recognition question and sample number limitation. Provided by the AI4Arctic competition, the preprocessed GCOM-W1 AMSR2 36.5GHz H polarization and Sentinel-1 SAR EW dual-polarization data, the CIS/DMI ice chart labels, and the pre-trained U-Net CNN model are employed to perform semantic segmentation of ice and water with
near-100% accuracy. Subsequently, within the U-Net semantically segmented ice mask, a multistage pixel-based ice detection algorithm developed on GLCM textures of SAR images and region growing approach, the Multi-textRG algorithm, refines the ice edge details. We validate the results on Landsat-8 and Sentinel-2 optical data yielding an overall accuracy of 83.11%, low false negative (FN) of 4.03% indicating underestimated low backscatter ice surfaces and higher false positive (FP) of 12.86% reflecting their resolution difference along edges. More importantly, we fused the SAR-based ice detection
with CIS/DMI ice charts and AMSR2 ASI SIC product obtaining SAR-Chart and SAR-AMSR2 labels, which enhance ice edge depictions and SIC variation contours. Repeating the two-step procedure with the high-precision SIC labels demonstrates the capability of U-Net model to extract detailed ice edges and stability of the Multi-textRG algorithm. The U-Net model trained on SAR-AMSR2 label achieves the highest $R^2$-score of 91.993%, the largest $OW_{recall}$ (recall of OW) of 99.268%, and large $ov40_{recall}$ (recall of ice with over-40% SIC) of 99.207%. Our algorithm framework solves the accurate
ice-water classification at all seasons and facilitates the sample labelling for improving SIC estimation accuracy based on CNN models.

## 1 Introduction

Synthetic Aperture Radar (SAR) data are vital for sea ice monitoring due to their high resolution, all-weather capability, and frequent observations. SAR-based sea ice classification algorithms have been explored with various approaches, initially
employing methods like simple thresholds, regression techniques, neural networks, and statistical models like Maximum



Likelihood and Bayesian classifiers. Recent advancements include the integration of support vector machines (SVM), random forests (RF), conditional random fields (CRF), and deep learning algorithms, especially convolutional neural networks (CNNs) (Leigh et al., 2014; Kortum et al., 2021). These methods leverage texture features, single or dual-polarization data, and sample labels from ice charts or manually drawn data. Zakhvatkina et al. (2019) and Song et al. (2021)
well review the SAR-based sea ice classification methods.

In this paper, previous sea ice classification algorithms are categorized into four groups. Firstly, Supervised pixel-based machine learning methods utilize texture features for classification, including SVM, RF, CRF and Multilayer Perceptron (MLP) methods that are trained on gray level co-occurrence matrix (GLCM) textures. These methods primarily rely on local texture features, making them highly sensitive to pixel-level details (Zakhvatkina et al., 2013; Zakhvatkina et al., 2017; Park
et al., 2020; Liu et al., 2015; Murashkin et al., 2018). Secondly, unsupervised block segmentations are combined with supervised pixel-based machine learning, such as the iterative region growing using semantics (IRGS) -SVM (i.e., MAGIC system) method (Leigh et al., 2014) and the IRGS-RF method (Jiang et al., 2022). The IRGS image segmentation and machine learning are conducted separately, and then the final ice-water labels classification is determined by incorporating an energy function (Leigh et al., 2014). One attempt at super-high resolution ice-water segmentation is by (Korosov and
Jeong-Won, 2016). The authors first use K-means segmentation in each 300×300 pixels block to obtain finely segmented small polygons of the whole Sentinel-1 SAR image, and then calculate the GLCM textures in each polygon (one polygon as one texture window) as the input to training SVM using manual labels on these polygons. Consequently, the identification of sea ice types heavily dependents on the segmented small size polygons.

In third category, statistical models are used as classifiers, like MSTA-CRF (Zhang et al., 2021), CRF combined with
CNN (Kortum et al., 2021) and clustering algorithms accounting for textural variations (Cristea et al., 2022; Doulgeris, 2015; Cristea et al., 2020). The fourth category features deep learning models directly training on SAR pixel intensities, employing convolutional structures to retrieve semantic context information (Song et al., 2018; Song et al., 2021; Boulze et al., 2020; Malmgren-Hansen et al., 2021; Wang and Li, 2020). Deep learning models are also used for SAR sea ice concentration (SIC) retrieval using Canadian Ice Service (CIS) charts or Advanced Microwave Scanning Radiometer 2 (AMSR2) SIC product as
labels (Wang et al., 2016; Wang et al., 2017b; Wang et al., 2017a; Cooke and Scott, 2019). Kucik and Stokholm (2023) well review the CNN algorithms developed on SAR images to mapping SIC or ice types. Despite advancements, these algorithms still face limitations in robustness.

The large interclass ambiguity between certain ice types and open water (OW) due to similar backscatter intensities and texture features in dual-polarized (e.g., HH and HV polarizations) SAR data, introduces challenges for the automatic sea ice
classification algorithms. Young ice growing in leads shows a wide range of HH intensities due to different micro-roughness and salinity levels, i.e., initially turning very bright (bright ice leads) and subsequently darkening to brightness levels similar to the surrounding ice (dark ice leads) in X-band and C-band SAR HH polarization (Guo et al., 2023). The bright young ice vs. heavily deformed ice (HDefI) and the dark young ice vs. deformed ice (DefI) respectively show largely overlapped histograms in SAR HH intensity and the GLCM textures (Guo et al., 2023), as well as in SAR HV images (Kortum et al.,



2021). Newly formed ice in OW, melt ponds and melted/rainy ice, can have the same dark backscatters as calm OW (Cristea et al., 2022; Niehaus et al., 2023; Zakhvatkina et al., 2017; Song et al., 2021). Winded OW instead is brighter than newly formed or wetted ice groups, and even more so than level first-year-ice (FYI) or less-rough FYI/multi-year-ice (MYI) (Boulze et al., 2020; Song et al., 2021). Specifically, melt-ponded ice with disk geometries shows a large contrast of intensity to the surrounding ice in SAR HV imagery (Sun et al., 2023). Complex surface characteristics (e.g., different

degrees of deformation, melting, geometric-roughness or micro-roughness, salinity, and wind speed) and radar modes (e.g., wavelength, polarization, and incident angle) are all important factors impacting SAR backscatters (Song et al., 2021; Lohse et al., 2020; Guo et al., 2023).

The large interclass ambiguity among similar ice types also exacerbate misclassification issues of the manual labels or ice charts (Park et al., 2020; Kortum et al., 2021; Song et al., 2021). When these similar types coexist in one scene, they may

be correctly distinguished with the semantic context textures and relative locations. However, when one of these similar types exists in one scene, it is very likely to be misidentified. Moreover, ice charts often include over-drawn sea ice edges with calm OW regions, impacting the parameter selection of machine learning methods. While methods like CRF and CNN can leverage semantic context information for better classification (Wang et al., 2017b), the accuracy and quantity limitations of labels may hinder the algorithm robustness. There is an urgent need for a larger dataset of accurate labels to

enhance algorithm performance.

In this study, we propose an algorithm framework to enhance sea ice observation and generate new sea ice concentration (SIC) labels for training potential CNN models. Unlike the typical unsupervised regional segmentation combined with supervised pixel-based learning, our approach prioritizes supervised global segmentation coupled with unsupervised pixel-based labeling. Initially, we conduct ice-water semantic segmentation using SAR and AMSR2 data inputs, the CIS or the

Danish meteorological institute (DMI) ice chart labels, and employing the U-Net CNN model provided by the AI4Arctic project. The AI4Arctic project, also known as the AutoICE challenge, offers a robust foundation for algorithm development, providing both raw and pre-processed training datasets and a pre-trained U-Net CNN model (Stokholm et al., 2022). Our methodology involves a two-step process. First, we utilize the U-Net model to distinguish between winded open water and thin ice/melted ice based on semantic segmentation. Subsequently, we develop a multilayer fine-recognition system called

the Multi-textRG algorithm using GLCM textures and regional growing to make ice and water pixel-level recognition in SAR images. This process is analogous to the multistage cloud recognitions in optical data that combining multiband image features to gradually separate several targets. In SAR images, dual-polarization of HH and HV are combined.

Dual-polarization observations are essential for improving SAR-based sea ice classification (Boulze et al., 2020; Zakhvatkina et al., 2019; Fors et al., 2016) or SIC estimations (Karvonen, 2014), owing to their sensitivity difference to wind

speed, incidence angle, and thin ice. Specifically, OW backscatters in HH-polarization exhibit distinct patterns under varying incidence angles, with being notably brighter than sea-ice at low angles and being typically larger than OW in HV-polarization in low to moderate wind speeds (Karvonen, 2014). Conversely, thin ice tends to have low backscatter values, approaching the noise equivalent sigma zero (NESZ), in HV polarization but not in HH polarization (Cristea et al., 2022).



Additionally, level FYI with smooth surfaces show rather higher backscatters in HV images than that of HH images. These
demonstrate the operational complementary surface features in HH and HV polarization for sea ice classification. However,
certain surfaces may not adhere strictly to these patterns, necessitating further experimentation for effective utilization of
dual-polarization data. At least, it has been observed that dual-polarized data are sufficient to achieve high-precision ice-
water recognition using machine learning algorithms (Kucik and Stokholm, 2023; Wang and Li, 2020).

The major error source for two opposite algorithm strategies, i.e., the second category and our algorithm, is the training
results from machine learning. While the neural networks aiming to achieve pixel-based classification with high precision
will be likely affected by the large ambiguity between local ice and local water, the neural networks using larger-scale
semantic context features to improve classification accuracy will inevitably lose original pixel resolution. Targeting high
precision in machine learning methods also requires a large amount of manual labels, which is time-consuming and labor-
intensive. In contrast, the advantage of our algorithm is the relatively high precision while retaining the benefit of high
resolution, without the need for more detailed manual labeling except ice charts. This may address the robustness limitations
in large-scale applications of SAR ice-water classification algorithms.

In this paper, we combined the U-Net model and the new proposed Multi-textRG algorithm to obtain detailed ice/water
pixels detection. We aim to achieve several goals: 1) to automatically perform ice-water classification with high resolution
and high precision, 2) to merge the ice pixel detections with CIS/DMI ice charts or AMSR2 ASI SIC product to produce two
new sets of SIC training labels with higher accuracies (respectively called SAR-Chart and SAR-AMSR2 SIC labels), and 3)
to explore the sensitivity of U-Net CNN model training on differently accurate SIC labels. The new labels may help for
developing better neural network models for SIC estimations. We also validated and compared the ice pixel detection results
from three U-Net models trained respectively on ice charts, SAR-Chart and SAR-AMSR2 SIC labels combined with the
unsupervised Multi-textRG algorithm ("3+1" experiment), to explore the stability of the Multi-textRG algorithm.

## 2 Data and processing

### 2.1 Sentinel-1 SAR data

We use the dual-polarization (HH and HV) Sentinel-1 extra wide (EW) mode Level-1 GRD data supplied by the
Sentinel-1A and 1B satellites operated by the European Space Agency. Each satellite carries an advanced C-band (5.405
GHz) SAR instrument to provide an all-weather, day-and-night supply of imagery of Earth's surface. The SAR has an
incidence angle of 20°-45°. The double orbits have a 6-day repeat cycle. The EW mode offers a 400 km swath width with a
20×40 m spatial resolution.

### 2.2 GCOM-W1 AMSR2 data

AMSR2 is a passive microwave (PM) instrument on board the GCOM-W1 satellite. The AMSR2 brightness temperature
(BT) data can be acquired during both day and night-time with a swath over 1450 km and a revisiting period of 2 days. The



level-1B AMSR2 product was normalized in values and matched with Sentinel-1 SAR images within the AI4Arctic competition datasets (further explained in Sect.2.3). The level-1B product stores the BT converted from the Level 1A antenna temperature using conversion coefficients. There are 7 bands (6.925/7.3, 10.65, 18.7, 23.8, 36.5 and 89.0 GHz) and 2 polarizations (H, V) of the AMSR2 level-1B data, with higher spatial resolution at higher frequencies. The channel specifications of AMSR2 data clarify the relative sensitivities of these bands to several ocean and atmosphere variables

(https://suzaku.eorc.jaxa.jp/ GCOM_W/wamsr2/whatsamsr2.html). Consequently, only 36.5GHz H polarization was selected as the combined input data due to its relatively higher resolution and lower sensitivity to winded water surface.

The AMSR2 BT data are also used for operational SIC estimations. The daily AMSR2 ASI SIC product provided by the University of Bremen has been used in a broad range of research applications to understand various aspects of Arctic sea ice dynamics and its interactions with the environment. We have compared the AMSR2 ASI SIC estimations with a SAR-based

sea ice extent product concerning their performance in depicting ice edges (Sun et al., 2023). To produce the SAR and AMSR2 fused SIC labels with higher accuracy, we selected the daily AMSR2 ASI SIC product obtained from the University of Bremen with a spatial resolution of 3.125 km.

### 2.3 AI4Arctic project dataset

The AI4Arctic project provides the raw dataset and the ready-to-train dataset. The competition was due on April 17,

2023. The newest dataset of version 2, posted on May 25, 2023, consists of 512 training and 20 test data files (links are on https://platform.ai4eo.eu/auto-ice/data). Sentinel-1 EW GRD data in HH and HV polarizations, AMSR2 Level-1B BT data in 7 bands and 2 polarizations (H, V), other auxiliary data (ERA5 weather data, distance-to-land information) and reference ice charts (including SIC, SOD and FLOE) produced by CIS and DMI are provided.

The difference between the raw and ready-to-train datasets is that the former includes the Sentinel-1 SAR data with an

alternative noise correction developed by the Nansen Environmental and Remote Sensing Center (NERSC) (Korosov et al., 2022), while the latter is a post-processed version of the raw dataset (with downsampling from 40 to 80 m pixel spacing, standard scaling, conversion of ice charts, removal of NAN values, mask alignment etc.). In the ready-to-train dataset, the SAR HH and HV backscatters, incidence angle, distance-to-land map and the ice charts are all resampled into 80 m pixel space resolution, with a matrix size of around 5000×5000. The other data are all resampled into 2 km grid resolution, co-

located and georeferenced to the geometries of Sentinel-1 SAR images.





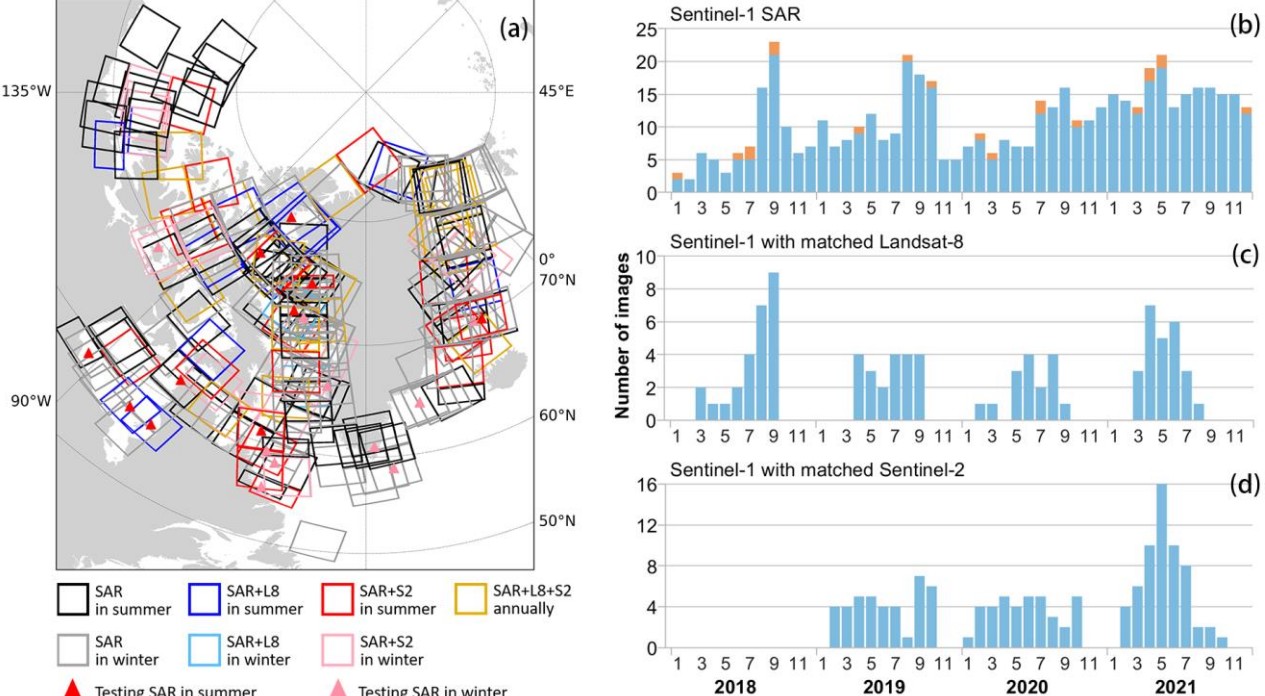

**Figure 1.** Data distribution. (a) shows the geolocations of 512 training images (boxes) and 20 testing images (triangles), which are differentiated with matched Landsat-8/Sentinel-2 (L8/S2) data or not and in summer (May to September) or winter (other months). (b) is the number distribution of all Sentinel-1 SAR images (blue: training, orange: testing) in month of the years 2018 and 2021. (c) and (d) respectively shows the number distribution of Sentinel-1 SAR images with matched Landsat-8 (85 scenes) and Sentinel-2 (134 scenes) images.

The AI4Arctic project datasets span the period from January 8, 2018 to December 21, 2021, with geographical coverage in the waters surrounding Greenland and the Canadian Arctic. Fig.1 shows the geolocations of SAR images marked into several types differentiated by matched Landsat-8/Sentinel-2 data or not, in winter or summer, and being testing data or not. The Landsat-8 and Sentinel-2 data are further introduced in Sect.2.5. In Fig. 1(a) and (b), the distribution of Sentinel-1 SAR images shows higher densities in the Greenland coast and the years 2020 and 2021 but uniform densities in winter and summer. The typicality of ice conditions of the AI4Arctic project datasets is also shown as the interpretation difficulty during the ice charts production from some SAR scenes even for professional ice analysts (Stokholm et al., 2022). This may be attributed to ambiguous surface features (e.g., melt ponds, smooth FYI, deformed FYI/MYI, newly formed ice and thin ice). More details can be found in the user manuals for the two datasets. Multi-series installments of articles related to AI4Arctic include (Esbensen, 2022; Stokholm et al., 2022; Stokholm et al., 2023; Kucik and Stokholm, 2023). In this paper, we used both the raw and ready-to-train datasets of the latest versions.



## 2.4 Sentinel-1 SAR data reprocessing

The complete processing of Sentinel-1 dual-polarization data should include geolocation interpolation, land masking, calibration, thermal noise removal, and additional incidence angle correction for HH images. The authors of the AI4Arctic project have processed and normalized the Sentinel-1 dual-polarization data for convenient training of expected CNN models.

Specifically, the thermal noise of Sentinel-1 EW mode HV-polarization images including the scalloping effect and sub-
swath banding effect is corrected in the AI4Arctic datasets using the state-of-the-art NERSC algorithm (Korosov et al., 2022). Based on the method proposed by (Park et al., 2018), the NERSC algorithm enhances the thermal noise removal procedure by adjusting the shape of the annotated range noise vectors provided by ESA to the shape proportional to the antenna range gain $G_{ar}$. Mismatch between the annotated noise vectors provided by ESA and the actual noise intensity shape in the range profile in each sub-swath has been early noticed in our experiments. The NERSC algorithm fundamentally
resolves this inaccuracy in thermal noise removal. However, only thermal noise is removed and textural noise still remains. Textural noise can roughen the GLCM textures on calm water surface, even on level ice surfaces without the presence of frost flowers on top (Park et al., 2019). Additional difficulties are thereby introduced for recognition of these and other low-backscatter ice types. The textural noise will also introduce large errors in the GLCM texture-based fine-recognition of ice pixels in the proposed Multi-textRG algorithm.

The open-source Python code for the NERSC algorithm and texture noise removal method is available, but it appears to be only compatible with the Linux system. Therefore, to use the state-of-the-art NERSC algorithm, we newly downloaded the zipped files of Sentinel-1 EW mode HH+HV polarization GRD data to obtain the DN values and the annotated calibration (referred to as $cali$) matrix in HV polarization. The $DN^2/cali^2$ represents the original backscatter values before noise removal. The AI4Arctic raw datasets provide the noise-removed backscatter values by the NERSC algorithm, which
should be considered as the $signal$. According to the calibration and denoising function provided by ESA, the $signal = DN^2/cali^2 - noise/cali^2$. $noise/cali^2$ is the calibrated noise, i.e., the noise equivalent sigma zero (NESZ). Thus NESZ $= DN^2/cali^2 - signal$. Subsequently, the textural noise was able to further removed using the algorithm in (Park et al., 2019). Negative values were retained after the texture noise removal to minimize the loss of textural details. Additionally, we observed that the AI4Arctic raw Sentinel-1 data with IPF version 2.8X is not processed with the azimuth noise removing.
Their azimuth noise ($noise_{az}$) was eliminated based on the function $signal = DN^2/cali^2 - noise_{az} * NESZ$. More knowledge can be found in (Park et al., 2018; Park et al., 2019; Sun and Li, 2021). Subsequently, GLCM textures were calculated based on the linear units of fully denoised SAR backscatter values. An image example is shown in Fig. S1 of Supplement S1 file.





## 2.5 Landsat-8 and Sentinel-2 validation data

Landsat-8 satellite and the Copernicus Sentinel-2A and 2B constellation are two important optical imaging systems. Landsat-8 level-2 and Sentinel-2 level-1C data both provide top-of-atmosphere (TOA) reflectance in the spectral bands of visible, near-infrared (NIR) and shortwave infrared (SWIR) ranges. Additionally, the two satellites provide a quality assessment (QA) band including snow/ice signs and cloud signs. The QA snow/ice signs in Landsat-8 and Sentinel-2 data are generated by strict cloud and snow detection algorithm sequences, see that of the Sentinel-2 Level-1C/2A product

(https://sentinels.copernicus.eu/web/sentinel/technical-guides/sentinel-2-msi/level-2a/algorithm-overview). Thus, we directly used the QA snow/ice signs excluding lands as "true" sea ice observations to validate our ice-water classification. We also used the TOA reflectance in natural color bands (red, green, blue) to visualize the optical features of sea surface, and the NIR and SWIR bands to manually correct the snow/ice detection.

    Fig. 1(a) shows the geolocations of the selected Landsat-8 and Sentinel-2 satellites, with the northernmost image

reaching the north coast of Greenland. However, useful scenes are often limited by prevalent cloud contamination. Fig. 1(c) and (d) show that 85 SAR images were matched with the processed Landsat-8 data and 134 SAR images were matched with the processed Sentinel-2 data. 54 of them overlaps on the same SAR images (yellow boxes in Fig. 1a). Additionally, the filtered Landsat-8 and Sentinel-2 data show an concentration on melting and early-freezing seasons.

    The Landsat-8 and the Sentinel-2 data were respectively selected from the collection 'LANDSAT/LC08/C02/ T1_L2' and

the collection 'COPERNICUS/S2_SR' overlapping each SAR scene within 1-day before or after the SAR imaging-time, merged by median calculation and cloud-masked by using the QA cloud signs, on the Google Earth Engine (GEE) platform. Then, the natural color bands, NIR, SWIR and QA snow/ice signs (totally 6 bands) of both Landsat-8 and Sentinel-2 are exported from the GEE platform by blocks with a resolution of 30 m. The block size is determined to improve export speeds on the GEE platform. All exported bands are resampled to SAR image geolocations. Evaluation metrics are then calculated

based on pixel-to-pixel validation (see Sect.3.4), and visualized comparisons for totally 219 matched images are provided in "valid_to_L8S2.zip" available at https://zenodo.org/records/10998517.

    Surfaces covered by clouds could be mistakenly identified as snow/ice or sea-water due to errors in cloud-mask information from the QA bands in Landsat-8 and Sentinel-2 data. Therefore, after the initial validation and visualization, we selected images with cloud-interfered errors in QA snow/ice to conduct further correction using the Modified Normalized

Difference Water Index (MNDWI) (Xu, 2007). The equation is given in following Eq. (1):

$$MNDWI = \frac{Green - SWIR}{Green + SWIR} \tag{1}$$

    Here, MNDWI is obtained by normalizing the Green band (band 3 in both Landsat-8 and Sentinel-2) and the SWIR band (band 7: 2.107~2.294 μm in Landsat-8 and band 12: 2202.4, 2185.7 nm in Sentinel-2). According to our experiments, MNDWI shows potential for recognizing water regions covered by clouds or cirrus while other NDWIs cannot. MNDWI

values are usually much lower in cloud- or cirrus-covered water regions. Meanwhile, MNDWI may also produce extremely



low values on some sea-ice surfaces. it does not exhibit stable value characteristics or stable thresholds for snow/ice and water classification. Therefore, we only used MNDWI for additional error correction combined with visual interpretation. Finally, the validation indexes for these images with cloud-interfered errors in QA snow/ice are re-calculated and re-visualized.

**3 Methods**

The proposed method includes two modules: a U-net CNN model used to extract the semantical segmented ice regions and a novel Multi-textRG algorithm to detect detailed ice pixels. They are sequentially explained in the following Sect.3.1 and Sect.3.2. The detailed SAR-based binary ice-water classification was then separately fused with the CIS/DMI ice chart and Bremen AMSR2 ASI SIC product to obtain two sets of new SIC labels as stated in Sect.3.3. The new SIC labels were

used to train the U-Net CNN model again and designed as control experiments in Table 1. Table 1 shows the structure and goals of this paper, including three complete experiments for detailed ice detections and two data-fused SIC samples. Section 3.4 then gives the evaluation metrics used to comparing our results with the Landsat-8 and Sentinel-2 QA ice/snow data.

**Table 1.** Structures of the experiments for U-Net training and Multi-textRG ice growing based on different SIC labels.

| # | Data | Methods | Results | Further processes or purposes | |
|---|------|---------|---------|-------------------------------|---|
| 1 | **Ice Chart SIC**, AMSR2, SAR | U-Net one + Multi-textRG algorithm | **Result one**: SAR growing ice | • fused with ice chart => **SAR-Chart SIC**.<br>• fused with AMSR2 ASI SIC => **SAR-AMSR2 SIC**.<br>• validated on the optical data. | |
| 2 | **SAR-Chart SIC**, AMSR2, SAR | U-Net two + Multi-textRG algorithm | **Result two**: SAR growing ice | Compared to **Result one**. | To explore:<br>**1)** Sensitivity of the U-Net model to details of SIC labels; |
| 3 | **SAR-AMSR2 SIC**, AMSR2, SAR | U-Net three + Multi-textRG algorithm | **Result three**: SAR growing ice | Compared to **Result one**. | **2)** Stability of the Multi-textRG algorithm to different semantic ice regions. |
| -- | SAR-Chart SIC | zipped into the AI4Arctic datasets (.nc files) | Additional SIC training labels. | | |
| -- | SAR-AMSR2 SIC | zipped into the AI4Arctic datasets (.nc files) | Additional SIC training labels. | | |






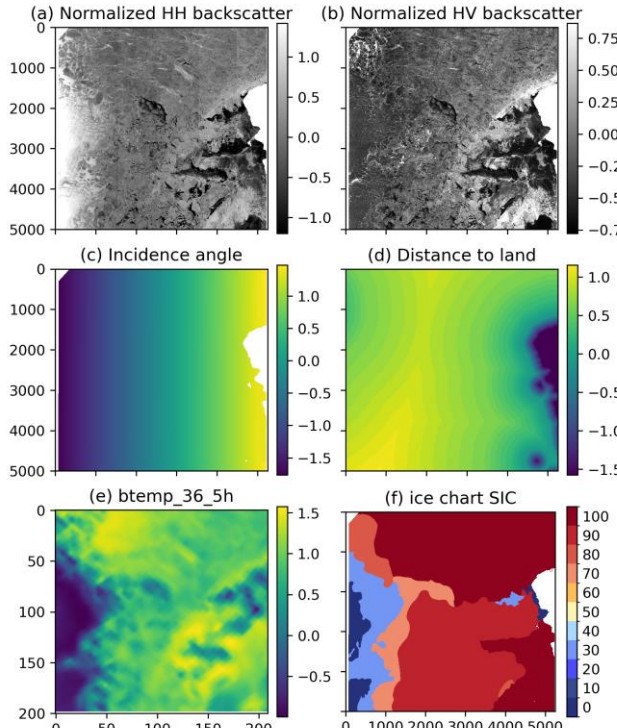

**Figure 2.** Example input features (a) - (d) and training label (f) of the U-Net one. In (e), the AMSR2 36.5GHz H-polarization image has approximately 200×200 pixels. It was subsequently resampled to the image size of approximately 5000×5000 pixels, consistent with all other inputs and outputs. This image was acquired on Apr. 29, 2021 in northeast Greenland, Fram Strait, same as that in Fig. 3 to Fig. 9.


**Table 2.** The experiment settings of three U-Net models used to get semantically segmented sea-ice region. The only difference for three models is the training labels of CIS/DMI chart SIC (see Fig. 2 and Fig. 6a), SAR-Chart fused SIC (Fig. 6b), and SAR-AMSR2 fused SIC (Fig. 6c).

| # | training images | validation images | testing images | *label | patch | batch | epoch | receptive field | levels | filters | SAR sigma unit | initial learning rate |
|---|---|---|---|---|---|---|---|---|---|---|---|---|
| **1** | 482 + repeated 19 | 30 + additive 3 | 20 | Chart SIC | 512² | 16 | 80 | 764 | 6 | 16, 5×32 | dB | 1e-4 to 5e-6 |
| **2** | 482 + repeated 19 | 30 + additive 3 | 20 | SAR-Chart SIC | 512² | 16 | 80 | 764 | 6 | 16, 5×32 | dB | 1e-4 to 5e-6 |
| **3** | 482 + repeated 19 | 30 + additive 3 | 20 | SAR-AMSR2 SIC | 512² | 16 | 80 | 764 | 6 | 16, 5×32 | dB | 1e-4 to 5e-6 |

* means that the only different setting for U-Net one, U-Net two and U-Net three is training labels.





### 3.1 Experimental settings of U-Net model

#### 3.1.1 U-Net one

Since the AI4Arctic project provides us great SAR-based training datasets including the professionally human-labelled ice charts, as well as a pre-trained U-Net model with pretty performance in SIC estimation (Kucik and Stokholm, 2023; Stokholm et al., 2022; Stokholm et al., 2023). We used the totally same U-Net CNN structure as provided in AI4Arctic competition (see figure 5 of paper https://doi.org/10.1109/TGRS.2022.3149323). Fig. 2 shows the example inputs and outputs of the U-net model. The normalized AMSR2 36.5GHz H-polarization data, the normalized Sentinel-1 dual-polarization (HH and HV) in dB unit, the Sentinel-1 incidence angles, and the pixel's distance-to-land map provided in the AI4Arctic ready-to-train dataset are used as the model input features (totally 5 input features). The SIC ice charts produced by DMI/CIS provided in the AI4Arctic ready-to-train dataset were used as training labels. The model inputs and reference labels are all resampled to 80 m pixel spacing during the model training. Table 2 then shows the basic experiment settings.

In this study, we refer to several conclusions in their experiments in (Kucik and Stokholm, 2023; Stokholm et al., 2022; Stokholm et al., 2023), including that 1) NERSC noise correction is superior for U-Net model predicting full sea ice covers with little variation in the SAR textures, 2) 11 classes of SIC shows the highest detection accuracies of water and 100% SIC, 3) the classification-based CrossEntropy (CE) loss function performs better at correctly predicting OW (0% SIC) and ice polygons of 100% SIC, and 4) using weighted loss functions result in a better average per class- and intermediate SIC, but with lower 0% and particularly the 100% accuracies. Consequently, we made the Sentinel-1 SAR data reprocessing as explained in Sect.2.4, kept 11 classes of SIC for model training rather than 2 classes of ice (over-0% SIC) and water (0% SIC), and used the classification-based CE function with solid loss weights—[0.05, 0.05, 0.13, 0.13, 0.13, 0.09, 0.09, 0.09, 0.09, 0.075, 0.075, 0.0], where 0.0 means the ignored mask class weight. The 11 class weights in 512 training scenes calculated by the method in (Stokholm et al., 2022) are [0.01, 0.55, 0.11, 0.14, 0.20, 0.12, 0.20, 0.11, 0.07, 0.06, 0.01] , [0.03, 0.63, 1.16, 1.35, 1.27, 0.86, 0.83, 0.64, 0.44, 0.15, 0.08], and [0.03, 0.82, 1.52, 1.51, 1.28, 0.67, 0.83, 0.70, 0.51, 0.26, 0.09], respectively for the ice charts, SAR-Chart and SAR-AMSR2 SIC labels. There are about four levels of the weights, whereas the large weights of 10% SIC (the second values: 0.55, 0.63, 0.82) were instead changed to 0.05, same as the 0% SIC, because of its high probability to be OW. The weights were manually set considering the balance between predicting 0% SIC and intermediate class of SIC. Additionally, we used 6 layers with a 764 receptive field to extract local image textures.

Besides, three different experiment settings were adopted during our model training. Firstly, we used the torch.optim. lr_scheduler.CosineAnnealingWarmRestarts() with the original learning rate of 1e-4, the iteration of solid 30 epochs and minimum learning rate of 5e-6 based on the Adam optimizer, which helps for accelerating the model convergence to the local minimum and then repeat exploring other local minimum within several periods.

Secondly, we redistributed the training and testing scenes to reduce the impact from sample imbalance between different ice types or ice conditions. Overall, 512 scenes of the AI4Arctic training dataset were divided into randomly selected 30 validation images and residual 482 training images, while 20 scenes of the AI4Arctic testing dataset were used for testing.



However, we found there is always a large error between the model prediction of highly winded water and level thin ice, whereas the scene number of two types occupy only small proportions in the AI4Arctic training dataset. Specially, large area of level thin ice or level FYI are found only in two scenes: '20190730T123155_cis_prep.nc' in training dataset and '20180623T114935_cis_prep.nc' in testing dataset. Training on the first scene has never accurately predicted the second scene. Thus, we used the latter within the training data list and used the former as one of the validation data. Moreover, 7 scenes of level thin ice were repeated 8 times and 12 scenes of winded open water were repeated 4 times within the AI4Arctic training data list, to increase the sampling frequency of two types and to finally achieve over-99% detection accuracies of them. Also, for effective validation, the 1 of the former and 2 of the latter were added to the validation data list. The repeated scene names are provided in Table S1 in Supplement S1 and the codes are available on https://zenodo.org/records/10973107.

Thirdly, we introduced the incidence angle matrix as one of the feature input, thus the cropped $512^2$-pixel patches are directly input to the model without any data augmentation. The models are trained for 80 epochs (approximately 40 h training duration) with the computer memory of two NVIDIA GeForce RTX 3090 24GB GPUs using PyTorch version 2.0.1 and cuda 11.8.

### 3.1.2 U-Net two and U-Net three

The experimental settings for U-Net two and U-Net three was totally same with that for U-Net one, except that the training labels were changed to be new SIC labels. Using the proposed method including the U-Net one and Multi-textRG algorithm, we acquired the growing ice-water classification result. Then, the binary ice-water image was merged respectively with the DMI/CIS ice charts and the AMSR2 ASI SIC products (see Fig. 6). The goal of two additional U-Net trainings is to explore if the model accuracy can be improved after changing to the SIC labels with higher-precision and higher-resolution in ice edges.

### 3.1.3 Model evaluation metrics

Model performance for 11 classes SIC estimation is assessed based on the statistical $R^2$ coefficient, as utilized and defined in (Stokholm et al., 2022). Inspired by the MacroBins metric proposed by (Stokholm et al., 2023), we further calculated separated recall metrics to select the best U-Net model for 2 classes ice-water segmentation.

$$OW_{recall} = \frac{water_{TP}}{water_{label}}, \quad ov40_{recall} = \frac{ice_{TP}}{ov40_{label}}, \quad bl40_{recall} = \frac{ice_{TP}}{bl40_{label}} \tag{2}$$

Here, $water_{TP}$ and $ice_{TP}$ mean the true positive pixel numbers of water (0% SIC) and ice (over-0% SIC), respectively. $water_{label}$, $ov40_{label}$ and $bl40_{label}$ then segment three regions, that is, open water: 0% SIC, large-concentration ice region: over-40% SIC, and low-concentration ice region: 0~40% (excluding 0%) SIC, according to the SIC classes in training labels.





In this study, we assumed the best U-Net model should detect the extents of 0% SIC and over 40% SIC labelled by ice charts, that are definitely OW and dense pack ice, with near-100% precision. Whereas the extents with 0~40% (excluding 0%) SIC labelled in ice charts are usually overestimated compared to SAR images. Thus, we selected the best-performing U-Net model for semantic ice segmentation with the highest sum of $R^2$, $OW_{recall}$ and $ov40_{recall}$ among 80 training epochs (see Fig. 7).

## 3.2 Multi-textRG algorithm

### 3.2.1 GLCM texture selection and computation

GLCM textures capture pixel-neighborhood correlation features within a defined window, akin to the super-pixel textures used in CRF (e.g., (Zhang et al., 2021)) and the semantic context information extracted by CNN (e.g., (Kortum et al., 2021)). These aspects are advantageous for distinguishing between ice types and open water (OW) with similar SAR intensities but differing relative positions to surrounding distinguishable ice/water pixels. GLCM textures are well analyzed for their inter-correlations and sensitivities in discriminating different sea ice types (Park et al., 2020; Murashkin et al., 2018; Guo et al., 2023). In this paper, we measured 13 GLCM textures that are calculated by using the code provided by NERSC with the mathematical expressions given by (Haralick et al., 1973). We also calculated 6 GLCM textures provided in (Li et al., 2021) with definitions from both (Haralick et al., 1973) (Conners and Harlow, 1980).

Our findings revealed that all 13 Haralick textures were sensitive to image noise, with significant correlation observed among several groups of textures (for details, refer to Fig. S2 in Supplement S1). This suggests that the 6 GLCM textures (including Sum Average, Energy, Entropy, Contrast, Correlation, and Homogeneity) are representative and sufficiently independent. Additionally, computing these latter 6 textures using MATLAB requires considerably less time and space compared to the former 13 textures calculated using Python codes. Consequently, we selected 3 textures from the 6, specifically Sum Average, Contrast, and Correlation textures as computed in (Li et al., 2021).

GLCM textures were computed using parameter settings of 64 gray levels, a window size of 32×32 pixels, slide step of 4 pixels, distance of 8 pixels, and orientations of 0°, 45°, 90°, and 135°. While Guo et al. (2023) investigated GLCM textures, they noted that textures derived from HH intensities in the logarithmic (dB) domain were influenced by a linear relationship with the incidence angle. Therefore, the 3 selected GLCM textures in both HH and HV polarizations were computed in the linear unit domain in our study.

### 3.2.2 Two additional GLCM texture combination

Selecting the three textures of Sum Avg, Cont, and Corr in both HH and HV polarizations enables us to explore limited combinations of them with multistage objectives, such as detecting various ice types with differing texture brightness. Initially, referring to Fig. 3, we defined goals across three stages: 1) identifying high backscatter ice in HH Cont or HV Cont (typically thick ice or bright ice leads), 2) recognizing moderate backscatter ice in HV Corr, and 3) detecting low/weak



backscatter ice in HH Corr (often thin or melted-surface ice). The primary challenge lies in the third step, particularly in scenes with HV Corr, where we must find ways to eliminate wind-disturbed water or enhance the contrast between water and

low backscatter ice as much as possible.

The ratio of HH to HV has been widely employed with demonstrated effectiveness for classifying sea ice types (as stated in our introduction). Therefore, we introduced the HV/HH ratio, which offers a distinct advantage: it reliably assigns near-zero values to certain waters under low incidence angles. This near-zero value ensures that any multiplication with other textures also yields values close to zero. Another significant advantage of HH images is that the backscatter values of thin

ice/melted surface ice are notably higher than the near-zero values of calm water in HH images, unlike the same zero backscatters observed in thin ice, melted surface ice, and calm water in HV images. However, using HH images necessitates the removal of the incidence angle effect.

Among the three GLCM textures in HH polarization, HH Cont transforms the extremely high backscatter of open water regions under low incidence angles (as seen in HH Sum Avg) into nearly zero texture values (Fig. 3). To leverage the

benefits of HH images, two additional textures are further processed with logarithmic normalizations: the ratio of HV to HH and the HH Cont.

$$rhvhh_{logn} = log_{10}\left(\left(\frac{HV\ Sum\ Avg}{HH\ Sum\ Avg}\right)^2 + 1.0\right) \tag{3}$$

$$hhCont_{logn} = log_{10}\left(\left(\frac{HH\ Cont}{200}\right)^{0.5} + 1.0\right) \tag{4}$$

The purpose of logarithmic normalizations is twofold: firstly, to stretch or compress low values to distinguish variously

bright ice surfaces (i.e., those with higher backscatter) from definitively dark water (i.e., areas with the lowest backscatter), and secondly, to compress non-zero values toward the higher end. Hence, the power values in Eqs. (3) and (4) are respectively set as 2 and 0.5 to ensure suitable stretching of the low values. The number 200 in Eq. (4) is utilized to linearly normalize HH Cont to a range typically spanning from 0 to 2. Ultimately, we obtained a total of eight useful textures, serving as the foundation for pixel-based ice-water classification.






**Figure 3.** Eight primary GLCM textures are used in the Multi-textRG algorithm. Among them, the (a) HH Sum Avg and (d) HV Sum Avg are the window-average intensities in HH and HV polarizations.

### 3.2.3 Procedure of Multi-textRG algorithm

The concept of multi-stage recognition becomes necessary when dealing with objects exhibiting low inter-class separation. This approach shares similarities with techniques such as cloud (shadow) masking and the one-vs-all method in machine learning. Despite the prevalence of neural network applications in sea ice classification, we have opted to revisit a more traditional empirical approach, considering the importance of generating reliable training labels automatically or semi-automatically. In unsupervised SAR ice-water classification algorithms, the available image information typically includes



pixel intensities and local texture features. Among these methods, threshold-separable features are particularly significant. Therefore, in this paper, the Multi_textRG algorithm is designed to leverage the regional textures of images and their associated local thresholds. This approach incorporates the classic GLCM textures and a simple threshold-based region growing method. The procedure involves experimental combinations of texture features, from simpler to more complex, layer by layer, to distinguish ice surfaces with varying backscatter levels.

Fig. 4 and 5 depict primary procedure and illustrate key steps of the Multi-textRG algorithm, respectively. The Multi-textRG algorithm accomplishes pixel-based ice detection in three stages by expanding upon three combined textures, which are:

$$Ctext_1 = hhSumAvg * hvCont * hhCont_{logn} * rhvhh_{logn} \tag{5}$$

$$Ctext_2 = hvCorr * (1 - hhCont_{logn}) * (1 - rhvhh_{logn}) \tag{6}$$

$$Ctext_3 = 1 - (1 - norm(hhCorr)) * hhCont_{logn} * rhvhh_{logn} \tag{7}$$

Here, the * denotes scalar multiplication and $norm()$ signifies scalar normalization to 0~1. The "1-" (i.e., 1 minus) and $norm()$ operations are used to concentrate the ice pixels in $Ctext_2$ and $Ctext_3$ on low values. Combining Fig. 4 with Fig. 5, the algorithm is presented as the following steps:

**1.** In the positive grayscale $Ctext_1$, non-black/whiter pixels represent ice, majorly thick ice or bright ice leads (Fig. 5a).
This brightness enables the determination of a global threshold (denoted as $t_1$) using Otsu's threshold method (Yuan et al., 2015) to segment the high backscatter ice.

**2.** In the negative grayscale $Ctext_2$, black pixels represent ice, while white pixels represent water (Fig. 5d). Comparing it to the HV Corr in Fig. 3, there is a significant amplification of contrast between ice and water, particularly in near-range (i.e., low incidence angle), making it possible to distinguish lower backscatter ice from the rough water surface. Another
feature is that ice intensities are largely compressed towards the lower end of histogram, facilitated by $hhCon_{logn}$ and $Rhvhh_{logn}$ textures. So that a simple region growing method first proposed by (Pal and Pal, 1993) is capable to segment moderate backscatter ice, as explained below in steps 3, 4, and 5. The second row in Fig. 5 illustrates the growing seeds and growing ice pixels based on $Ctext_2$.

**3.** Selecting ice seeds in each sliding window (300×300 pixels): A first-peak with triangle threshold method is used to
determine individual thresholds based on the intensity histogram of the U-Net segmented ice pixels. In Fig. 4, the histograms in the first dotted box illustrate two typical threshold approaches. First, a trough appears after the first peak in the histogram, a right triangle is constructed using the horizontal and vertical axis values of the first peak and the trough. The threshold $t$ is then selected at the intersection of the histogram fit-line and the hypotenuse midline of the triangle. Second, if there is no trough after the first peak in the histogram, twice "triangle hypotenuse midline threshold"
are used to determine the threshold. The global threshold matrixes, $t_2$ and $t_3$, are acquired for $Ctext_2$ and $Ctext_3$.



4. Selecting growing thresholds in each sliding window (300×300 pixels): The individual growing threshold is selected based on the average absolute intensity difference within a 5-pixel step of the ice seeds in each window, resulting in a threshold matrix $t_4$.

Note: The sliding window of 300×300 pixels is achieved with a 50-pixel step.

5. Growing new ice seeds in each sliding sub-window (61×61 pixels): The sub-window centered at $(i, j)$ with a square radius of 30 pixels is moved by every 5-pixel step. Taking the average intensity of all seeds in the 61×61 sub-window as $'a'$, any pixel in the sub-window with intensity $'b'$ meeting the condition of $b - a < t_4$ is recognized as a new ice seed. The expanded ice seeds will be used for ice growth in the next sliding sub-window. Once all sub-windows have

completed movement, the ice pixels will grow to their maximum extent based on the threshold matrix $t_4$.

6. In the negative grayscale $Ctext_3$, black pixels represent ice while white pixels represent water (Fig. 5g). Comparing it to $Ctext_2$, the contrast between low backscatter ice surfaces and OW is further amplified, particularly in the disk-shaped ice regions (maybe melt ponds). Region growing, including steps 3, 4, and 5, is conducted once more. The third row in Fig. 5 shows the growing seeds and growing ice pixels based on $Ctext_3$.

7. Finally, the binary ice detections from the 3 stages are combined in Fig. 5(j). Comparing it to the U-Net predicted semantic ice regions, the ice boundary details are well extracted.

The running time includes round 16s for U-Net prediction on two NVIDIA GeForce RTX 3090 24GB GPUs, about 270s for SAR image processing, about 684s for GLCM textures calculation, and about 540s for Multi-textRG ice detection based on a computer with an Intel(R) Core(TM) i5-10400F CPU @ 2.90GHz and 64GB of RAM.








**Figure 4.** The procedure of Multi-textRG algorithm. The $Ctext_1$ is used to detect high backscatter ice with the Otsu's threshold (signed as $t_1$). The $Ctext_2$ and $Ctext_3$ are sequentially used to detect moderate backscatter ice and low backscatter ice, using the region growing(RG) method. The RG method consists of two parts, i.e., 1) selecting ice seeds based on thresholds (signed as $t$, $t_2$, $t_3$) calculated in each sliding window of 300×300 pixels using a "first-peak with triangle threshold" method (illustrated in the first dashed box) and 2) expanding new ice seeds using RG method (illustrated in the second dashed box).






**Figure 5.** The illustrations of 3 combined textures (the first column) for 3-stage-ice-detection in the Multi_textRG algorithm. (f) shows the U-Net segmented semantic ice region (see Fig. 7d), from which ice edges are extracted and marked as brown contours in (d), (e), (h) and (i). Then, (j) shows the final union growing ice by "logical AND" operation of (b), (e) and (i). In $Ctext_1$, the whiter pixels are ice; in $Ctext_2$ and $Ctext_3$, the blacker pixels are ice.

## 3.3 Data fused SIC labels

The detailed SAR binary ice-water classification, i.e., the SAR growing ice pixels in Fig. 5, was then separately fused with the CIS/DMI ice chart and Bremen AMSR2 ASI SIC product to obtain two sets of new SIC labels. The fusion of SAR and AMSR2 ice data includes the following steps: 1) Calculating the average within the sliding window of 15×15 pixels to transform the SAR binary ice-water matrix to SAR SIC values. 2) Clipping and georegistering the daily Arctic 3.125 km grids AMSR2 ASI SIC product to individual SAR images.3) Calculating average of the SAR SIC and the processed AMSR2 ASI SIC values on each pixel. 4) Using the function round((fused SIC*1000)/100) to acquire 0-10 integral SAR-AMSR2 fused SIC labels. Since the step 2) for CIS/DMI ice charts has been performed in the AI4Arctic dataset. The fusion of SAR results and ice charts includes the above steps 1), 3), and 4).





Two examples are shown in Fig. 6, consistent with the cases in Fig. 8 and Fig. 10. Compared with the ice chart SIC, the SAR-Chart SIC maintains similar class segmentations but introduces more details in low SIC regions. Firstly, see Fig. 6(a) and 6(b), the 30% SIC polygon in Chart SIC labels is replaced by two segmentations of higher 60% and lower 10%-20% SICs in SAR-Chart SIC labels. The large area of 30% SIC polygon in Fig. 6(a) may cause confusion to the U-Net model that OW and broken ice floes share the same SIC values. Now the higher 60% and lower 10%-20% SIC values direct the U-Net

model to differentiate OW and broken ice distinctly, thus potentially improving the precision of the U-Net model. Moreover, the below 80% SIC polygons in Fig. 6(a) increased to be 90% in Fig. 6(b). This is because the GLCM textures include local window information and the Multi-textRG algorithm tends to identify dense pack ice regions as near-100% SIC. This highlights a certain limitation of the method first performing SAR texture-based ice-water classification and then converting it into percentage SIC values.

Compared with the former two SICs, the SAR-AMSR2 SIC shows higher precision and more detailed spatial distribution. In Fig. 6(c), the SAR-AMSR2 SIC includes 11 SIC classes with the pattern of 100% SIC smoothly reducing to 0% SIC from compact ice to ice edges, particularly displaying more reasonable labels in over 60% SIC regions. The 50% values in SAR-AMSR2 SIC labels mean the complement effect by the fusion of two data sources. The second case, Fig. 6(d), (e) and (f), shows similar changes of SIC labels except for some SIC reductions over the probably wet ice surface, which will be

discussed in Sect.4.2.2.

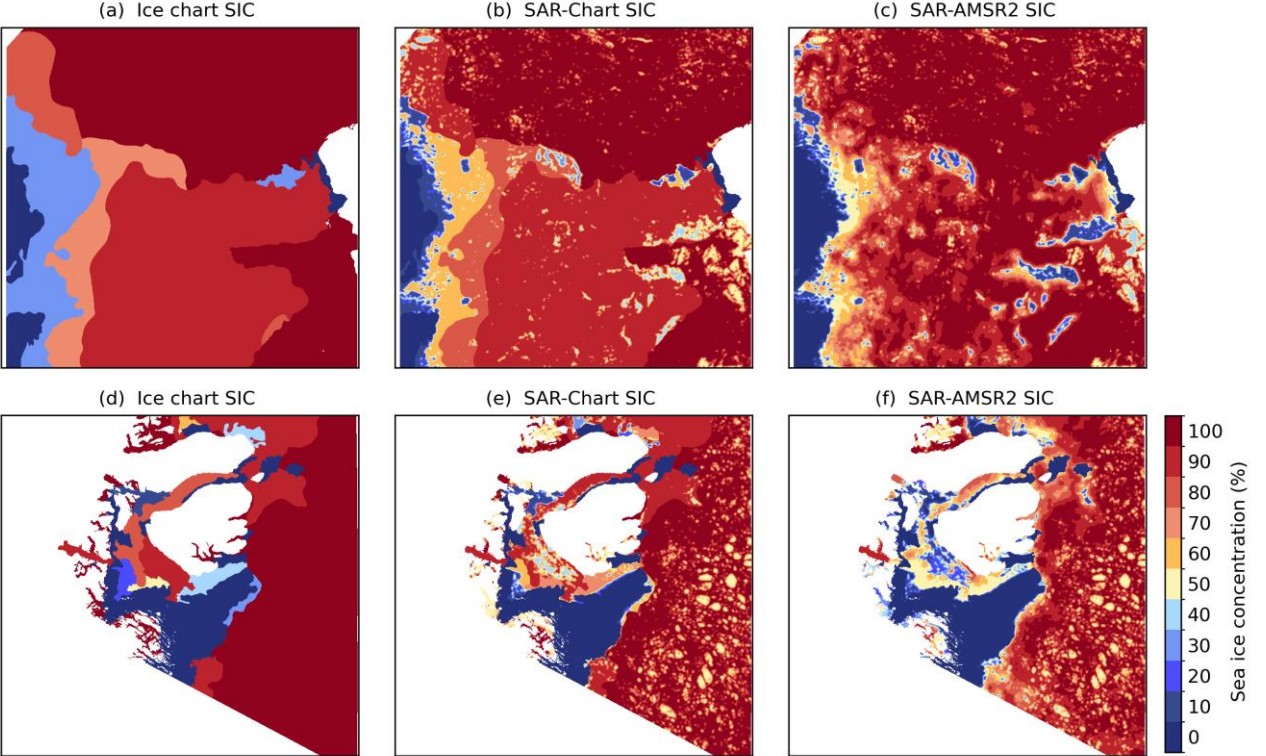

**Figure 6.** Two example images displaying the Chart SIC, SAR-Chart fused SIC and SAR-AMSR2 fused SIC labels.



**3.4 Binary ice/water validation metrics**

The validation metrics comparing with Landsat-8 and Sentinel-2 data include overall accuracy (OA: sum of TP and true
negative (TN)), false negative (FN: ice in Landsat-8/Sentinel-2 and OW in SAR) rate, false positive (FP: OW in Landsat-8/Sentinel-2 and ice in SAR) rate, and $Parea$ (area proportion of unmasked effective pixels). They are calculated based on the pixel-to-pixel validation:

$$OA = \frac{\{ICE_{l8s2} \text{ and } ICE_{sar}\}_{np}}{unmask_{np}} + \frac{\{OW_{l8s2} \text{ and } OW_{sar}\}_{np}}{unmask_{np}} \tag{8}$$

$$FN = \frac{\{ICE_{l8s2} \text{ and } OW_{sar}\}_{np}}{unmask_{np}} \tag{9}$$


$$FP = \frac{\{OW_{l8s2} \text{ and } ICE_{sar}\}_{np}}{unmask_{np}} \tag{10}$$

$$Parea = \frac{unmask_{np}}{unmask_{np} + mask_{np}} \tag{11}$$

The subscript $np$ is the number of pixels. $mask_{np}$ means the pixel number of individual SAR or Landsat-8/Sentinel-2 image masked as nan-values (including land mask, cloud mask, etc.), while $unmask_{np}$ represents the other regions. $Parea$ is used as the weights for calculating the average OA values among totally 225 SAR images with matched Landsat-
8/Sentinel-2 images. Then $ICE_*$ and $OW_*$ are respectively ice and open water pixels, where the subscript $sar$ means the Multi-textRG ice-water classification in SAR images and the subscript $l8s2$ means the QA snow/ice sign in Landsat-8/Sentinel-2 images.

**4 Results and discussions**

**4.1 U-Net model results**

Fig. 7(a), (b) and (c) show the training loss, $R^2$ and the recall values of three U-Net models (U-Net one, U-Net two, and U-Net three). Since the period of the learning rate scheduler of warm restart is 30 epochs, those metrics show their decrease or increase trends with the disturbances consistent to the warm restart period. By selecting the epoch with the largest sum of $R^2$-score, $OW_{recall}$ and $ov40_{recall}$ values, the best-performing U-Net models were selected respectively at epoch 56, 29, and 75 (marked as red font).

Fig. 7(d), (e) and (f) show the predicted SIC images of three U-Net models, with poorly identification of intermediate classes of SIC but well detected ice edges. The U-Net one predicted the largest ice area with SIC over 0% and SIC of 30%, the U-Net two predicted the lowest ice area with SIC below 50%, while the U-Net three predicted the largest ice area with SIC of 40% to 50%. We have tried to reduce the sample imbalance between different ice conditions during the experiment settings, however, the inaccurate polygon labeling or the imbalanced proportions of 11 SIC classes still cause great difficulty




for predicting the intermediate classes of SIC. In Table 3, only slight improvement of $R^2$ values occur in U-Net one to U-Net three when results are validated to three distinct training samples. Therefore, additional validations of the predicted SIC values against SAR-AMSR2 SIC labels are presented in Table 4. Suppose that SAR-AMSR2 SIC is the closest to ground truth, disregarding the U-Net three are trained on SAR-AMSR2 SIC itself, significant improvement is found on the $R^2$ values. It is evident that the development of more accurate U-Net SIC prediction necessitates training on balanced samples with finer-grained labels.

The U-Net model predicts ice edges in higher precision as the SAR-Chart and SAR-AMSR2 SIC labels include highly differentiated classes and class spatial distributions, see the predicted SIC maps and the evaluation metrics as well. The increased $OW_{recall}$ and the decreased $bl40_{recall}$ in Table 3 and 4 from U-Net one to U-Net three state that, as accurate ice edges are included in labels, the U-net predicted ice edges also shrink to the truth. The ice edges predicted by U-Net three shows the highest precision reflected by the over-99% $OW_{recall}$ and $bl40_{recall}$ values and the lowest $bl40_{recall}$. Referring to Fig. 7(f), the U-Net three predicted ice extent seems to no longer need pixel ice detection using the Multi-textRG algorithm. Certainly, the first-round result from U-net one combined the Multi-textRG algorithm is still necessary. Though additional experiments are not given out, the SAR HH, HV polarizations and AMSR2 36.5GHz H polarization were found as necessary for accurate ice-water classification task.

On the other hand, numerous studies have shown that the classification challenges majorly exist in the wind-driven OW and the low backscatter ice surfaces including level thin ice and wet ice surfaces (melt ponds). Thin ice and wet ice surfaces are typically observed within dense pack ice regions with over-40% SIC. Conversely, brighter ice surfaces including brash ice, frazil ice and newly formed ice often occur within the low SIC regions of 0%−40%. In Table 3, the globally over-97% $OW_{recall}$ and $ov40_{recall}$ values demonstrate the proficient capabilities of U-Net model to identifying these similar ice surfaces within wind-driven OW and over-40% SIC regions. Thereby, the Multi-textRG algorithm is able to achieve precise pixel-based ice detection without the disturbance from winded OW.



**Figure 7.** (a), (b), (c) show the training losses and evaluation metrics of three U-Net models. (d), (e), (f) show the predicted SIC map by three U-Net models. The red numbers mark the optimal U-Net models selected with the largest $ov40_{recall}$ within the first 50% $OW_{recall}$.


**Table 3.** The evaluation metrics of three U-Net models (**# 1, 2, 3**) validated to three distinct SIC labels (see Fig. 6). The $R^2$ is calculated for 10-class SIC predictions. $OW_{recall}$, $ov40_{recall}$, and $bl40_{recall}$ are calculated for 2-class ice or water.

| # | labels | 30 validation scenes validated to respective labels | | | | 20 testing scenes validated to respective labels | | | |
|---|---|---|---|---|---|---|---|---|---|
| | | validation accuracy: $R^2$ | $OW_{recall}$ | $ov40_{recall}$ | $bl40_{recall}$ | testing accuracy: $R^2$ | $OW_{recall}$ | $ov40_{recall}$ | $bl40_{recall}$ |
| 1 | Chart SIC | 88.503% | **98.337%** | **98.547%** | 70.979% | 87.066% | **97.343%** | **98.993%** | 79.789% |
| 2 | SAR-Chart SIC | 93.464% | **99.448%** | **98.604%** | 55.486% | 88.705% | **99.064%** | **99.235%** | 63.275% |
| 3 | SAR-AMSR2 SIC | **94.279%** | 99.484% | 99.000% | 60.512% | **91.993%** | 99.268% | 99.207% | 57.889% |





**Table 4.** The evaluation metrics of three U-Net models (# 1, 2, 3) validated to SAR-AMSR2 SIC labels.

| # | labels | 512 training and validation scenes validated to **SAR-AMSR2 SIC** | | | | 20 testing scenes validated to **SAR-AMSR2 SIC** | | | |
|---|---|---|---|---|---|---|---|---|---|
| | | *validation accuracy: $R^2$ | $OW_{recall}$ | $ov40_{recall}$ | $bl40_{recall}$ | testing accuracy: $R^2$ | $OW_{recall}$ | $ov40_{recall}$ | $bl40_{recall}$ |
| 1 | Chart SIC | 68.200% | **96.404%** | **99.626%** | 90.065% | 82.912% | **91.647%** | **99.610%** | 91.872% |
| 2 | SAR-Chart SIC | 63.418% | **98.006%** | **99.375%** | 82.733% | 80.764% | **95.763%** | **99.541%** | 84.483% |
| 3 | SAR-AMSR2 SIC | **74.763%** | **99.439%** | **99.385%** | **60.864%** | **91.993%** | **99.268%** | **99.207%** | **57.889%** |

* marks the $R^2$ values of 512 training and validation images are calculated by first discarding the negative (nonsense) $R^2$ value of each individual image and then averaging the rest. The $R^2$ values of 20 or 30 scenes are calculated by all pixels at once.

## 4.2 Case study of the Multi-textRG algorithm results

### 4.2.1 Thin ice

Using U-Net one and the Multi-textRG algorithm, we obtained the SAR growing ice---ice-water classification results. Figures 8 and 9 present a case verification of SAR growing ice respectively against Landsat-8 and Sentinel-2 optical data. The mosaic Landsat-8 and Sentinel-2 data with effective values (excluding cloud masks) were exported from the GEE platform in segmented blocks (red grids in the figures). The two optical datasets cover an overlapping region located in the upper right section of the SAR image. Sentinel-2 data, in particular, covers a large area of thin ice with very low backscatter

values, which has consistently posed challenges for SAR-based sea ice classification methods. As a result, the validation metrics of Fig. 8 show an OA of 92.8%, FN of 5.07%, FP of 2.17%, and $Parea$ of 9.7%, whereas the validation results of Fig. 9 yield a lower OA of 70.4%, higher FN of 5.08%, substantially higher FP of 24.6%, and $Parea$ of 16.9%. The visual interpretation of the optical and SAR images provides several key insights and discussions.

Firstly, dark point and linear regions are visible within the red boxes in the SAR HV gray image (Fig. 8b), while most of

them are not excessively dark (i.e., not the darkest) in SAR HH gray image (Fig. 8a). These regions show as white or gray (means the ice) in the Landsat-8 and Sentinel-2 natural color images (Fig. 8d and 9d). The comparison between dual-polarization SAR and optical images indicates the possibility of these regions representing thin ice in dark ice leads. The ratio of HH to HV backscatters shows high values over the newly formed thin ice (Fig. 8a and 8b), which underscores a crucial physical mechanism for surface scattering. Rougher surfaces can result in a greater proportion of incoherent return

waves (Beaven et al., 1994)( Woodhouse, 2006), while smooth surfaces are instead dominated by coherent scattering with a limited depolarization effect (Lievens et al., 2022). However, extremely low ratio of HV to HH occurs at some of thin ice (Fig. 3g), thus the Multi-textRG algorithm identified them as water pixels (Fig. 8e).



Secondly, a greater challenge lies in identifying larger areas of thin ice with both polygon shape and texture features similar to OW, illustrated by the low backscatter regions within and around the blue box region marked in Fig. 9(a). The
pure white appearance in optical images and low backscatter intensities in SAR images suggest that these are smooth thin ice surfaces covered with dry snow (Fig. 9). This image was captured on April 29, 2021 on the north side of the Fram Strait (80°N to 82°N), near the end of winter close to the North Pole, which shows the probability for presence of dry snow. In optical images, reflection is the primary mechanism for wavelengths below 3 μm (König et al., 2001). The reflectance/albedo of snow in the visible band is mainly influenced by impurities, snow grain size, and liquid water content that can increase the
effective grain size and thus reduce the albedo of the snowpack (König et al., 2001). The blue or light gray surfaces in the upper right area of Fig. 8(d) and 9(d) represent wet snow or ice, while the lower right area of Fig. 9(d) appears completely white indicating dry snow. C band co-polarized microwaves can penetrate dry or refrozen snow to depths ranging from meters to tens of meters (Rott and Nagler, 1993) thus get the backscatter of the snow-ice interface. However, C band cross-polarization may experience enhanced signal depolarization due to scattering within the dense anisotropic snow volume at a
certain snow depth (Lievens et al., 2022). The low backscatter in HV polarization (Fig. 9b) provides exact evidence that the region comprises thin and dry snow with a weak signal depolarization effect.

The HH and HV polarizations both receive low backscatter values from the smooth snow-ice interface under near-specular scattering at high incidence angles (around 38°-46°). However, smooth thin ice surface and small snow depth doubly increase the coherent scattering and decrease incoherent scattering. HH polarization typically receives higher returns
than HV polarization making smooth thin ice and OW more distinguishable in HH polarization than in HV polarization. In other words, smooth thin ice appears brighter than the darkest OW in HH image, while both appear equally darkest in HV image. Additionally, the presence of irregular cross-linear bright textures, likely resulting from strong directional scattering due to significant surface roughness (ice deformations), aids in identifying thin ice surfaces from OW in SAR HH images (see Fig. 5(g) and Fig. 9(f)).

The Multi-textRG algorithm effectively identifies most of the snow-covered thin ice regions in Fig. 9 by leveraging a region growing method and complementary backscatter features in dual-polarizations. However, due to the limitation of C-band SAR observation that smooth thin ice could have the darkest ice surfaces in both HH and HV polarizations, the ice regions circled in blue lines in Fig. 9(f) and 9(g) were not recognized. Moreover, these snow-covered thin ice surfaces seem to be smoother (without any dark dots) than the surrounding ice even in the Sentinel-2 RGB-to-grayscale image (Fig. 9g). By
visual interpretation on the SAR HH and HV images, the whitest ice region within the blue box marked in Fig. 9(a) should be MYI. The topographic difference between MYI and level thin ice attributes to the varied smoothness in Fig. 9(g).





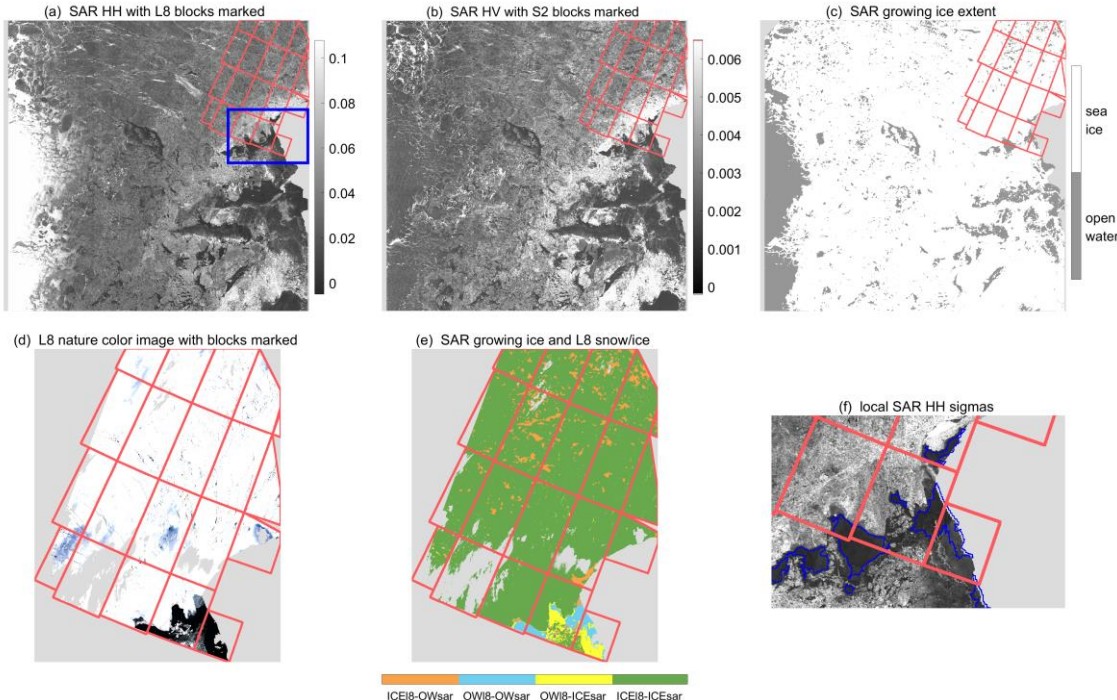

**Figure 8.** Case one: thin ice, validating the SAR growing ice resulted by U-Net one plus Multi-textRG algorithm to Landsat-8 (L8) data. (d) and (e) depict the view narrowed down to the effective values region of L8. (f) zooms in on the local SAR HH image within the blue box region. The red grids denote the locations of the optical images, and the blue contours in (f) are the SAR growing ice edges, and the non-values are masked as light gray color, same with that in Fig. 9 and 10.

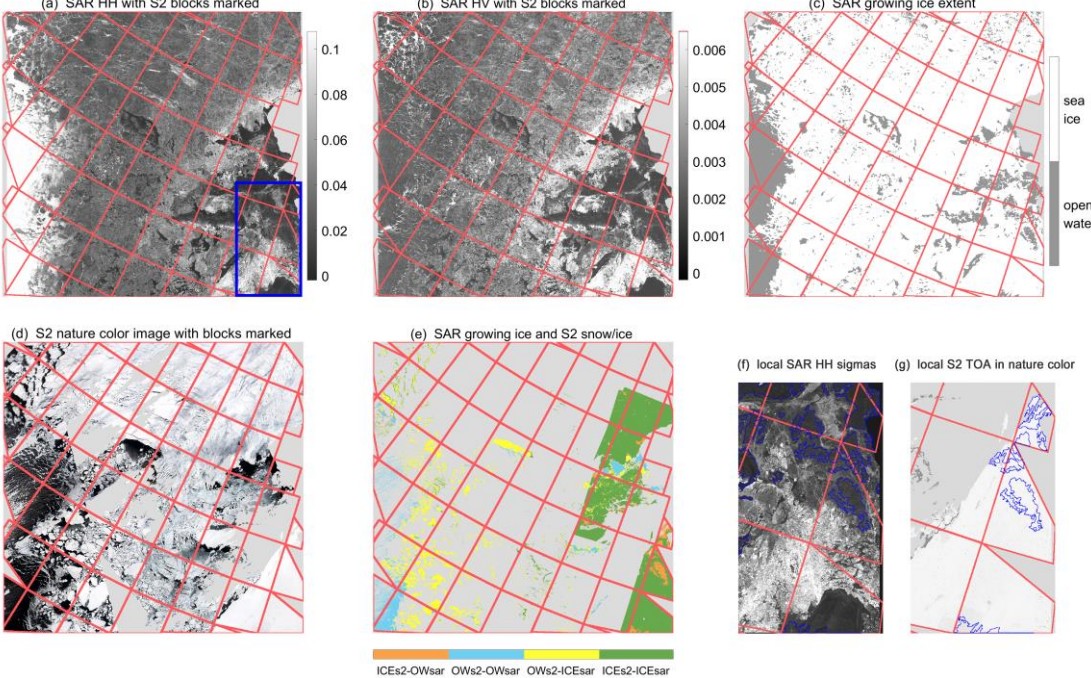



**Figure 9.** Case one: thin ice, validating the SAR growing ice resulted by U-Net one plus Multi-textRG algorithm to Sentinel-2 (S2) data. The area with QA snow/ice signs (e) is typically smaller than that with positive QA cloud-mask signs (d) in S2 data. (f) zooms in on the local SAR HH image and (g) focuses on the local S2 natural color image, within the blue box region.

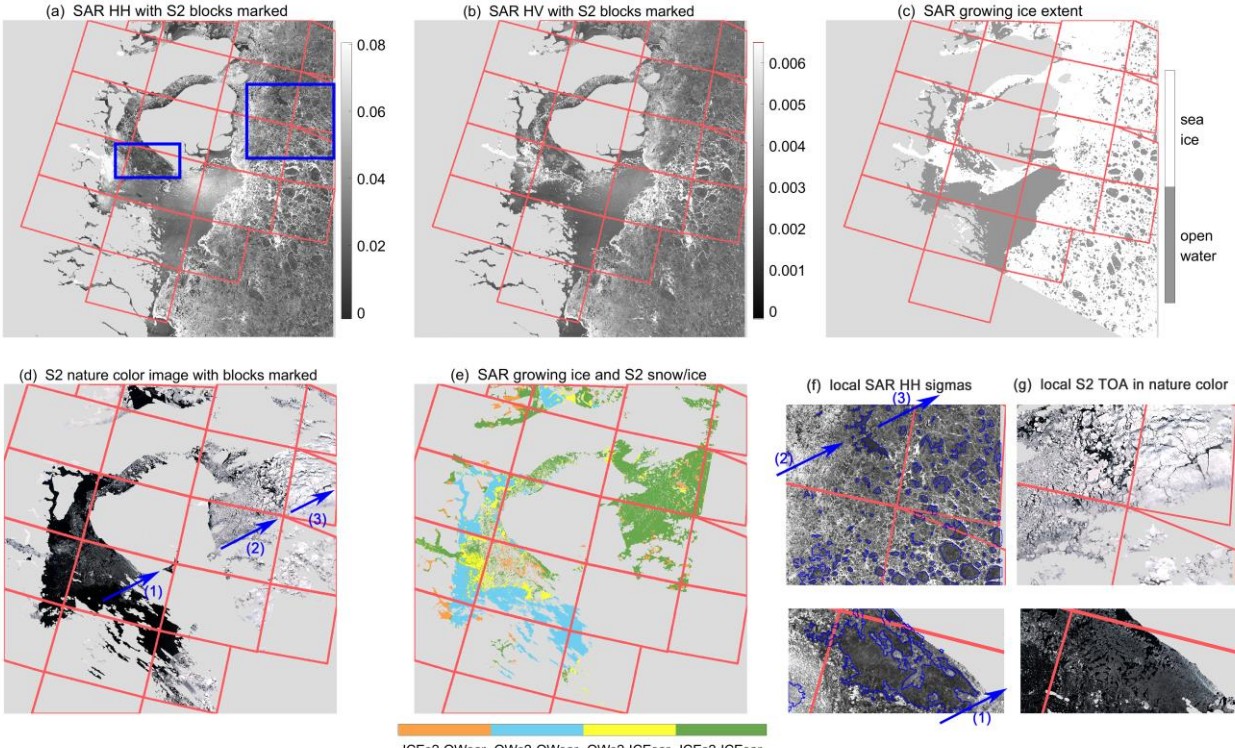

**Figure 10.** Case two: melted ice surfaces, validating the SAR growing ice resulted by U-Net one plus Multi-textRG algorithm to Sentinel-2 (S2) data. (f) zooms in on the local SAR HH image and (g) focuses on the local S2 natural color image, within the two blue boxes. Three arrows display the supposed offshore wind direction.

### 4.2.2 Melted ice surfaces

Figure 10 presents another case study. Low backscatter ice within or surrounding the blue boxes and wind-driven water surfaces at the image center are distinguishable in both SAR HH and HV images. Consequently, our algorithm successfully identifies the wind-driven open water, but largely fails to recognize the disk-shaped ice regions with low backscatters. In Sentinel-2 natural color image, notably increased albedos on ice surfaces are observed along the direction indicated by the three blue arrows, corresponding to decreased meltwater content. However, the ice surfaces in the SAR HH and HV images within the large blue box (i.e., the second and third arrow segments) exhibit similar grayscale and texture features. This can be attributed to strong offshore winds increasing surface melting upstream and driving snow accumulation until the third arrow segment, over which radar microwave penetrates the snow getting returns from snow-ice interface. The albedo difference is thereby not reflected on the SAR HH images over the second and third arrow segments. Fig. 10 has the OA value of 81.2%, the FN value of 9.19%, and the FP value of 9.59%.





The three arrow segments correspond to different ice conditions. Referring to Fig. 8(d) and 9(d), the low grayscale values in the first arrow segment in Fig. 10(d) suggest the presence of large proportions of meltwater on these ice surfaces or within the ice volume. Meltwater can decrease both optical albedos and radar backscatter. Thus, Fig. 10(f) also displays mottled darkness in the first arrow segment. In the second arrow segment, dense pack ice is observed with dark water gaps visible in the higher-resolution Sentinel-2 image in Fig. 10(g), whereas most of these water gaps are not visible in the SAR HH images due to the coarser pixel resolution. Conversely, numerous disk-shaped black ice regions with similarly low grayscale values as those in the first arrow segment are found in the second and third arrow segments, indicating the presence of meltwater over the ice surfaces as well. Comparing Fig. 10(f) and 10(g), it can be inferred that the disk-shaped water accumulates on different sizes of pack ice and is surrounded by dense bright ice leads with newly formed ice. These low backscatter disks may be melt ponds or wetted snow covers. Discriminating between them is challenging because the sea ice in the optical image has undergone certain ice motions, and there are no melt pond features visible on the optical ice surfaces.

The Multi-textRG algorithm fails to identify 100% of these black ice surfaces within semantic ice regions segmented by the U-Net. Except for the bare information from SAR C-band observation on them (as discussed in Sect.4.2.1), another reason is that the third step of the Multi-textRG algorithm assumes there must be water pixels shown as the darkest grayscale within the U-Net segmented ice regions. This is considering that limited area of OW within the U-Net segmented ice regions prohibits forming a sufficiently rough water surface thus showing extremely low backscatters. Only increasing the texture calculation window can help to derive enough features for discerning the black ice polygons from water surface. For example, the widths of black ice within the blue box in Fig. 9(a) and the small blue box in Fig. 10(a) are up to 1000*1000 pixels in 40 m resolution, making it challenging for the GLCM textures in a sliding window of 32*32 pixels to capture enough contextual information to identify the black ice accurately. However, using larger-scale features is typically the responsibility of U-Net and other machine learning models, which may struggle to detect ice details while maintaining high precision. Despite this, the good news is that the U-Net model trained on SAR-chart and SAR-AMSR2 fused SIC labels still successfully identifies all thin ice (Fig. 7e and 7f). This indicates it is sufficient to use the Multi-textRG algorithm to extract training samples.

The ratio of HH to HV backscatters also shows a high sensitivity to wind-driven water surfaces under near to middle incidence angles. In two cases, the bright water surfaces in HH image are all well discerned. However, some overestimation of the wind-driven SAR growing ice edges still occurs due to the limited pixel resolution of GLCM textures and fuzzy region growing. The sharp sigma gradient at ice edges can cause high HV Cont texture values in at least one window width of 32*32 pixels in 40 m resolution away from the ice edges. The OW in at least one window width is therefore easier to be wrongly identified as sea ice. The distance between the detected ice edge and the true ice edge depends on the window size and sliding distance parameters set during GLCM texture calculation. Fig. 8(f) shows a certain distance of the blue OW contours from white sea ice edges, with uneven distance values along the ice edges partly attributed to the region growing window size of 61*61 pixels instead of 1-by-1 pixel. This means that the inner ice seeds will grow across the peak texture values at ice edges and extend to a certain width (below 30 pixels) of water regions. Comparing the yellow area in Fig. 9(e)



to 9(d) and in Fig. 10(e) to 10(d), some wind-driven newly formed ice (probably frazil ice) are wrongly masked as cloud, while the Multi-textRG algorithm overestimates SAR ice extent along these broken ice edges. Caused by the difference in resolutions (160 m for SAR growing ice and 30 m for Sentinel-2 image) as well, high FP values of 25.6% and 9.59% were respectively acquired in Fig. 9 and Fig. 10.

Nevertheless, the experiential insights into ice and water gained from the visual interpretation of SAR and optical images
were instrumental in refining the Multi-textRG algorithm.

## 4.3 Stability of the Multi-textRG algorithm

Fig. 11(a), (b), and (c) present scatter statistical values for three validation metrics of experiment one, combining U-Net One with the Multi-textRG algorithm. Compared with 219 Sentinel-2 or Landsat-8 images, the SAR growing ice demonstrates overall weighted averages of OA as 83.11%, FN as 4.03%, and FP as 12.86% by *Parea*. The low FN value
indicates a small area of underestimated low backscatter ice surfaces with meltwater or smooth thin ice, highlighting the strength of the Multi-textRG algorithm. Conversely, the higher FP value primarily reflects overestimated ice regions along ice edges due to the coarser resolution and fuzzy region growing in SAR images, as discussed in the case studies. This overestimation suggests that the Multi-textRG algorithm may lose some accuracy in describing ice edge details at times. While this has minimal impact on ship navigation, it might affect heat flux calculations, as the overestimated windy water
surface has a much higher heat flux than ice surfaces. On the other hand, these errors are contained in the model training of U-Net two and U-Net three, resulting in more nuanced semantic ice segmentations obtained solely through U-Net model training (see Fig. 7). Furthermore, Fig. 11(d) shows the ice area differences between results two and three to result one (see Table 1). The median ice detection differences are respectively 44140 and 55022 pixels, which well illustrate the stability of the Multi-textRG algorithm based on semantic ice segmentation in different degrees of details.

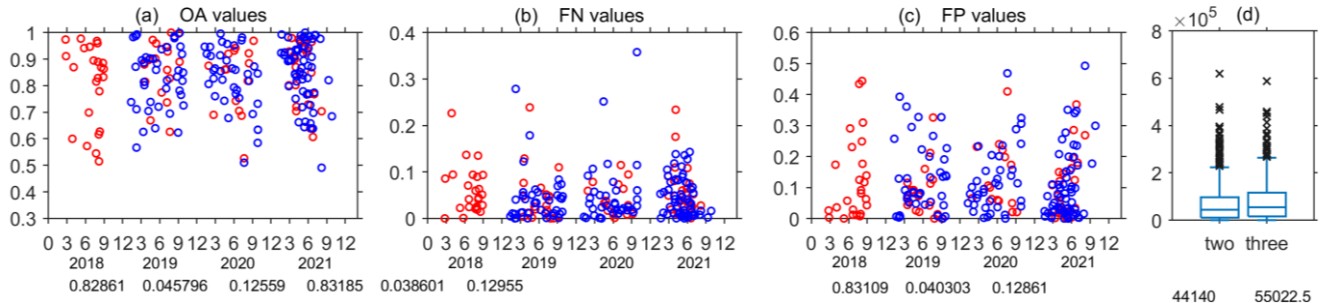


**Figure 11.** (a), (b) and (c) show the scatter values of the OA, FN and FP evaluation metrics for the SAR growing ice extent produced by U-Net One combined with the Multi-textRG algorithm, compared with Landsat-8 (red circles) and Sentinel-2 (blue circles) datasets. (d) shows the boxplots illustrating the ice area differences between results two and result three relative to result one across 532 SAR images.



## 5 Conclusions

This study proposes a novel algorithm framework that combines a supervised U-Net model with an unsupervised Multi-textRG algorithm based on the AI4Arctic competition datasets, to achieve fine ice-water classification and to acquire more precise SAR-involved/fused SIC labels. Two types of labels are generated: SAR-Chart SIC labels, obtained by fusing SAR ice-water classification results with CIS/DMI ice charts provided within the AI4Arctic datasets, and SAR-AMSR2 SIC labels, obtained by fusing SAR ice-water classification results with the AMSR2 ASI SIC product provided by the University

of Bremen. To explore the sensitivity of U-Net model to CIS/DMI ice charts and two new SIC labels, three experiments (named U-Net one, two and three) with the same model settings are conducted and evaluated by using $R^2$ and targeted recall values. Subsequently, the semantic ice segmentations output from three U-Nets are used for detailed ice pixel detection, from which the results demonstrated the stability of the Multi-textRG algorithm. Case studies and OA, FN, FP metrics are validated and calculated based on Landsat-8 and Sentinel-2 optical data. Conclusions are given below in three aspects: the

Multi-textRG algorithm, labels, and the U-Net model.

Identifying certain ice types presents challenges, including winded OW, level thin ice, newly formed ice, and melted ice surfaces (wet snow or melt ponds). Our method framework aims to address these challenges firstly by using neural networks for precise identification of complex ice surfaces and varied backscatter water surfaces in SAR images, and secondly by means of experiential knowledge and threshold-based region growing to achieve multi-stage ice detections akin to cloud

mask processing in optical images. During the Multi-textRG procedure, winded OW no longer poses difficulties in identifying ice surfaces at various grayscale levels. However, extremely low backscatters of smooth thin ice or melted ice surfaces still prove difficult for their pixel-level recognition. The overall average FP value suggests limited underestimation of low backscatter ice, at approximately 4.03%. Conversely, overestimated ice regions along ice edges caused by resolution difference between SAR and optical data, GLCM texture window size, and region growing window size, are examined to be

as high as approximately 12.86%, resulting in a lower OA value of 83.11%. Nonetheless, these errors do not propagate in the U-Net model, as indicated by the output results of the models trained on SAR-Chart and SAR-AMSR2 SIC labels.

Furthermore, the incorporation of two combined GLCM textures—logarithmic normalization of the ratio of HV to HH and logarithmic normalization of HH Cont—further enhances ice-water contrast, highlighting the complementary effects of dual-polarization on ice information extraction. Compared with completely unsupervised segmentation, the ice detection

using the Multi-textRG algorithm is controllable through manual parameter adjustments, such as GLCM texture calculation and region growing thresholds. Inter-comparison of the Multi-textRG algorithm results in three experiments demonstrates great stability among different semantic ice masks. However, the robustness of the algorithm under parent settings still needs verification over larger areas. Nevertheless, the semi-automated procedure has the potential to significantly alleviate the manual workload associated with ice chart production or sample labeling.

Compared to ice charts, SAR-Chart SIC labels reveal additional classes, small areas of water, and accurate ice edges differentiated by lower SIC one side and higher SIC on the other within polygons below 40% SIC. While SAR-AMSR2 SIC



labels achieve the merged ice extent beneficial from SAR-based marginal ice zone and AMSR2-based thin ice observations, boasting the highest accuracy in class spatial distribution despite some SIC reductions appearing over melted ice surfaces. Except for SAR dual-polarization data, AMSR2 36.5GHz H polarization data are found necessary for accurate identification

of winded open water. The inter-comparison of three U-Net model outputs underscores its high sensitivity to the detail and precision of training labels. Notably, U-net three achieves impressive metrics, including an $R^2$-score of 91.993%, an $OW_{recall}$ of 99.268% and an $ov40_{recall}$ of 99.207%, indicating its superior performance with SAR-AMSR2 SIC labels. Moreover, U-Net three demonstrates precise ice edge detection closely aligning with Multi-textRG algorithm delineations, as evidenced by the lowest $bl40_{recall}$ of 57.889%. More importantly, sample balance is crucial to the predict performance of

neural network models, especially under the various sea ice conditions in SAR images. It is necessary to balance the sample numbers of level FYI and winded open water, so that near-100% accuracy of ice water classification was finally achieved by the U-Net model in our experiments.

To summarize, our algorithm framework combines a CNN model and an empirical algorithm utilizing ice charts, SAR dual-polarization GRD data, and AMSR2 36.5GHz H-polarized data as inputs, and enables generating large batch of fine

ice-water classification products and high-precision training labels. The model training employing data-fused SIC labels demonstrates that the relatively simple structure of U-Net model is fully capable of semantical ice-water classification in high precision and in large-scale operational production.

**6 Future works**

For further accurate pixel-level recognition of low backscatter ice, including thin ice or melted ice, we propose two

potential directions for improvement. Firstly, we suggest continuing SAR-based targeted detection of ice types. This involves extracting all small "non-ice" polygons within the identified ice extent by the Multi-textRG algorithm, labeling them as melt ponds, level thin ice, or water gaps (ice leads), expanding their boundaries regionally to encompass the surrounding sharp-bright edges, and then employing CNN models to identify them by learning local textures and surrounding ice edge shapes. Secondly, using the valuable information from optical or radiometer data is recommended.

Three surface types can be easily recognized by visual interpretation in optical images, while thin ice has been reported to be accurately identified using AMSR2 89 GHz data because of its shallow penetration depth and resulting insensitivity to ice thickness (Ivanova et al., 2015).

Alternative models could substitute the U-Net CNN model. Improving the accuracy of ice-water classification with CNN models necessitates higher sample granularity or precision. Our U-Net experiments demonstrate successful ice-water

classification, whereas achieving accurate multi-category SIC prediction remains a challenge. Kucik and Stokholm (2023) combined the ice discriminator (IceDisc) and intermediate discriminator (InetDisc) to improve the performance for intermediate classes however under the precision limitation of human labels. Improved SIC predictions are expected by using the SAR-Chart or SAR-AMSR2 fused SIC labels.

Further experiments across the Arctic Sea are necessary to test the robustness of the algorithm framework. This
framework can serve as a reference for fine ice type classification or other classification studies. The objective of sea ice
information extraction is to achieve more accurate classification of ice-water, identification of ice surface characteristic
parameters, and potential SIC estimation, thereby enhancing our understanding of the role of polar sea ice in the Earth
system.

**Code and data availability**

Codes and testing data are available at https://zenodo.org/records/10973107. The visual validations of our ice-water
classification results to Sentinel-1 and Landsat-8 optical data are available at https://zenodo.org/records/10998517. The data-
fused SAR-Chart and SAR-AMSR2 SIC labels are available at https://zenodo.org/records/10973107 and https://zenodo.org/
records/10974340.

**Author contributions**

The methodology, experiments, analysis and manuscript writing were implemented and carried out by Yan Sun. Experiment
result analysis, experiment redesign, validation and manuscript content organization were developed in collaboration with
Shaoyin Wang, Teng Li, Chong Liu, Yufang Ye, Xi Zhao. The study was carried out under the supervision and support of
Xiao Cheng. All authors have reviewed the manuscript.

**Competing interests**

The authors declare that they have no conflict of interest.

**Acknowledgements**

This study was supported by the National Natural Science Foundation of China (Grant Number: 41925027, 42206249), and
the Innovation Group Project of Southern Marine Science and Engineering Guangdong Laboratory (Zhuhai) (Grant Number:
311023008). The authors would like to thank the AI4Arctic team who provided the ready-to-train dataset and the well-
designed U-Net model (https://platform.ai4eo.eu/auto-ice/data). We also greatly thank the European Space Agency (ESA)
for providing Sentinel-1 and Sentinel-2 satellite data; the University of Bremen, Bremen, Germany for providing the
AMSR2 ASI SIC product; and the United States Geological Survey (USGS) for providing Landsat-8 satellite data. The
insightful comments from the reviewers and editors are highly acknowledged.



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
