# Peer review of "Combining the U-Net model and a Multi-textRG algorithm for fine SAR ice-water classification"

_EGUsphere, 2024_

## Referee Comment (RC1)

**Review of the manuscript "Combining the U-Net model and a Multi-textRG algorithm for fine SAR ice-water classification" by Yan Sun, Shaoyin Wang, Xiao Cheng, Teng Li, Chong Liu, Yufang Ye, Xi Zhao**

The manuscript presents a complex set of experiments with U-net convolutional neural networks, texture features and fusion techniques aiming at predicting accurate sea ice concentration retrieval from SAR and PMW data. Despite several novel ideas contained in the manuscript I would recommend rejecting it for two major reasons.

**The first reason** can be summarized as follows: the ice charts are used as input to the algorithm and the output of the algorithm is validated against the same ice charts. It is quite difficult to understand this process from the description in the manuscript (see major reason 2) but I can briefly present the algorithm to explain my concern:
1. The "U-net one" and the "Multi-textRG" algorithm process S1 SAR data and generate "SAR growing ice" product.
2. The "SAR growing ice" is fused with the AI4Arctic ice charts to produce "SAR-Chart SIC".
3. "SAR-Chart SIC" is validated against the AI4Arctic ice charts.

It may seem counterintuitive, but in this chain of algorithms, a worse behavior of step 1 would lead to a better validation metrics on step 3. In the extreme case of total failure of U-net and Multi-textRG which produce zero concentration everywhere, the $R^2$ metric would be 100% because the output of step 2 would be almost the same (biased) input ice chart!

Continuation of the processing chain presented in the manuscript (i.e., U-net two, and three, etc.) only escalates this mistake: now the CNNs are trained and validated on the outputs of the first CNN. In the same extreme case of total failure of step 1 above, it should be very easy for "U-net two" to predict only zeros. No surprise the accuracy reaches 99%…

It is not an uncommon mistake to blend in target labels into input data in the ML oriented research, nevertheless it must be avoided. The only way is to find good reference data that allows fair evaluation of the novel / high resolution / high precision / high accuracy, etc. algorithms. Such data must be of the same quality as the expected results, with the same resolution / precision / informativeness, etc. And it must be substantial and representative to draw a statistically solid conclusion.

**The second reason** is that this paper is very difficult to read. Partly because it is too long and too many different algorithms are blended together, partly because the narration suffers from many logical mistakes, partly because of the language.

To overcome these problems I would suggest the following.
If this material makes it to a journal, it should be split at least in two logically separable parts: 1. The Multi-textRG algorithm (in fact, U-net + Multi-textRG but with major focus on Multi-textRG); 2. Sensitivity of U-net on level of details in input labels (U-net two, three, etc.). The first manuscript must be evaluated against accurate independent data,

and the second one can be a theoretical examination with synthetic data – "How would a U-net behave if we had higher precision data?"

For improving the logic of narration, it is good to keep similar style paragraphs and sections together and not to jump back and forth. For example, the paragraph on line 93 – 104 should be moved before the statement of the goal of this work (e.g., at line 81); sections 3.1.3 and 3.4 can be merged as they present validation methodology; Eqs. 5 – 7 can be grouped with Eqs. 3, 4 as they present new texture features for classification. Explanations of algorithms can also be improved for a better readability, for example the table 1 should be rearranged into a flow chart; the steps of Mutitextrg algorithm on page 16 and 17 should be rewritten in a similar style with a clear indication what is on input, what is the method, what is the results (e.g., Step 1: A global threshold (t1) is determined using Otsus' method on all values of Ctext1. The threshold is used to split all pixels into two groups: thick ice or bright ice leads. [*This is my interpretation of a cryptic description of Step 1 in the manuscript, it may be wrong...*]).

For improving the language, I can recommend avoiding jargon and consistently using the same terms. For example, "growing ice" means to me an increase of sea ice thickness due to freezing. If you mean "growing ice labels" please use that or define your abbreviation.